

# The effect of marine aggregate parameterisations on global biogeochemical model performance

Daniela Niemeyer[1], Iris Kriest[1], Andreas Oschlies[1]

[1]Helmholtz-Zentrum für Ozeanforschung Kiel (GEOMAR), Düsternbrooker Weg 20, 24105 Kiel, Germany

*Correspondence to: Daniela Niemeyer (dniemeyer@geomar.de)*

**Abstract.** Particle aggregation determines the particle flux length scale, possibly the marine oxygen concentration and thus affecting the volume of oxygen minimum zones (OMZs) that are of special relevance for ocean nutrient cycles and marine ecosystems, and that have been found to expand faster than can be explained by current state-of-the-art models. To investigate the impact of particle aggregation on global model performance, we carried out a sensitivity study with different

parameterisations of marine aggregates and two different model resolutions. Model performance was investigated with respect to global nutrient and oxygen concentrations, as well as extent and location of OMZs. Results show that including an aggregation model improves the representation of OMZs. Moreover, we found that besides a fine spatial resolution of the model grid, the consideration of porous particles, an intermediate to high particle sinking speed and a moderate to high stickiness improve the model fit to both, global distributions of dissolved inorganic tracers and regional patterns of OMZs,

compared to a model without aggregation. Our model results therefore suggest that improvements not only in the model physics, but also in the description of particle aggregation processes can play a substantial role in improving the representation of dissolved inorganic tracers and OMZs on a global scale. However, dissolved inorganic tracers are apparently not sufficient for a global model calibration, which could necessitate global model calibration against a global observational dataset of marine organic particles.

**1 Introduction**

Oxygen is – beside light and nutrients - fundamental for marine organisms, such as bacteria, zooplankton, and fish. Only few, specialised groups can tolerate regions of low oxygen, commonly referred as oxygen minimum zones (OMZs). These regions are located in the tropical upwelling regions, where nutrient rich water enhances primary production and subsequent transport of organic matter to deeper waters, which triggers respiration and consumes oxygen. Together with weak

ventilation (which supplies oxygen), this results in oxygen concentrations well below 100 mmol m$^{-3}$. Global models that are used to reproduce OMZ's volume and location, and their evolution under climate change, differ with respect to the biogeochemical parameterisations as well as with respect to physics (Cabré et al., 2015), resulting in disagreements between projected OMZ extent (Cocco et al., 2013). To date, it is not clear, if these differences can be attributed to the differences in the model's biogeochemistry or the physical model.

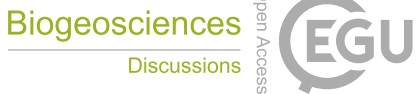



One potential parameter affecting distributions of dissolved oxygen and thereby the volume and location of OMZs is the biological carbon pump (Volk and Hoffert, 1985). Global ocean model studies show that the biological pump is important for the distribution of dissolved inorganic tracers in the ocean (Kwon and Primeau, 2006, 2008), as well as atmospheric $pCO_2$ (Kwon et al., 2009; Roth et al., 2014). It further affects the feeding of deep sea organisms (Kiko et al., 2017) as well as

the OMZs volume (Kriest and Oschlies, 2015). The biological carbon pump can be subdivided into three components: production of organic matter and biominerals in the euphotic surface layer, particle export into the ocean interior and finally their decomposition on the sea floor (Le Moigne et al., 2013). Estimates of the export of organic carbon out of the surface layer range from 5 to 20 Gt C yr$^{-1}$, with the large uncertainty illustrating the gap in our understanding of this process (Henson et al., 2011; Honjo et al., 2008; Keller et al., 2012; Laws et al., 2000; Oschlies, 2001). Further uncertainties are associated

with the exact shape of the particle flux profile (e.g. exponential function vs. power law; Banse, 1990; Berelson, 2002; Boyd and Trull, 2007; Buesseler et al., 2007; Lutz et al., 2002; Martin et al., 1987) and its possible variations in space and time. Recent studies suggest conflicting evidence with regard to the spatial variation of the particle flux length scale (Guidi et al., 2015; Marsay et al., 2015). Also, the underlying mechanisms for a potential spatio-temporal variation remain unclear: some studies attribute this to variations in temperature and associated temperature-dependent variation in remineralisation (Marsay

et al., 2015), while other studies derive this from variations in particle size distributions (Guidi et al., 2015).

One mechanism, that leads to a variation in particle size distribution, consists in the formation of marine aggregates, which exhibit variable sinking speeds. For example, Alldredge and Gotschalk (1988) and Nowald et al. (2009) found sinking rates for aggregates ranging between 10 and 386 m d$^{-1}$. Particle sinking speed, and thus the particle flux profile, depends on mineral ballast (Armstrong et al., 2002; Ploug et al., 2008), porosity and particle size (Alldredge and Gotschalk, 1988;

Kriest, 2002; Smayda, 1970). Large particles are associated with high sinking speed and fast passage through the water column, resulting in low remineralisation and thus a small OMZs volume, and vice versa. It can therefore be expected that particle aggregation favouring fast sinking speeds can alter the volume of OMZs compared to small particles with low sinking speeds (Kriest and Oschlies, 2015).

However, there are still some gaps in our understanding of the parameters that control the aggregation rate as well as the

particles sinking behaviour. For example, in-situ measurements show almost no dependency between diameter and sinking speed (Alldredge and Gotschalk, 1988) whereas aggregates produced on a roller table show a noticeable relationship (Engel and Schartau, 1999). Furthermore, values for stickiness, which defines the probability that after collision two particles stick together, vary over a wide range. Stickiness depends on the chemistry of the particle's surface (Metcalfe et al., 2006) and the particle type (e.g. Hansen and Kiørboe, 1997) and ranges between almost zero and one (e.g. Alldredge and McGillivary,

1991; Kiørboe et al., 1990). Thus, aggregation as one process that induces variations in particle size, and thus sinking speed, is only loosely constrained through its parameters.

To explore these relationships further and to examine whether a spatially variable sinking speed improves the fit of a global biogeochemical model to global distributions of dissolved inorganic tracers and regional patterns of OMZs, this study uses the three-dimensional Model of Oceanic Pelagic Stoichiometry (Kriest and Oschlies, 2015), coupled with a module for





particle aggregation and size-dependent sinking (Kriest, 2002). Given the large uncertainty associated with parameterisations of marine aggregates, we carried out 36 sensitivity experiments, in which we varied parameters relevant for particle aggregation and sinking. As in previous studies, the model`s fitness is evaluated by the Root Mean Square Error (RMSE) against observational data of dissolved inorganic tracers, namely $PO_4$, $NO_3$ and $O_2$ (Kriest et al., 2017). This study

additionally determines the model fitness with respect to extent and location of OMZs, following the approach by Cabré et al. (2015).

To examine the above-mentioned questions, and explore the effects and uncertainties of a model that simulates particle dynamics on a global scale for a seasonally cycling stationary ocean circulation, our main questions are as follows:

1. Does a model that includes explicit particle dynamics improve the representation of observed $PO_4$, $NO_3$ and $O_2$?

2. Does a model that includes explicit particle dynamics improve the representation of observed OMZs, and do the 'best' parameters with respect to this metric agree with those constrained by dissolved inorganic tracers?

3. What are the effects of uncertainties in the parameterisation of organic aggregates on model results?

4. Can the assumptions inherent in the model confirm either of the spatial particle flux length scale maps proposed by

Marsay et al. (2015) or Henson et al. (2015)?

This paper is organised as follows: we first describe the model and its assessment with regard to dissolved inorganic tracers and OMZs, including the sensitivity experiments carried out with the model. We then present the outcome of the sensitivity experiments, with special focus on the metrics defined above. We finally examine and discuss derived maps of particle flux

length scales against the background of maps derived from observed quantities (Henson et al., 2015; Marsay et al., 2015).

## 2 Model Description and Methods

### 2.1 Oceanic transport

In this study, we used the 'Transport Matrix Method' (TMM) (Khatiwala et al., 2005), as an efficient offline method to simulate biogeochemical tracer transport with monthly mean transport matrices (TMs). Additional fields of monthly mean

wind, temperature and salinity extracted from the underlying circulation model are used to simulate air-sea gas exchange of oxygen, and to parameterise temperature-dependent growth of phytoplankton. For our experiments, we used two different types of TMs and forcing fields: one set derived from a coarser resolution (hereafter called MIT2.8), and one from a finer resolution version, based on a data-assimilated circulation (ECCO1.0) (Stammer et al., 2004). The MIT2.8 forcing and transport represent a resolution of 2.8° x 2.8° and 15 depth layers with a thickness ranging between 50 m and 690 m.

ECCO1.0 TMs and forcing are based on a resolution of 1° x 1° and 23 depth layers, with a thickness ranging between 10 m and 500 m. Further details about the two setups can be found in Kriest and Oschlies (2013).



In general, we used a time step length of 1/2 day for physical transport, and a time step length of 1/16 day for biogeochemical interactions in the coarse resolution, MIT2.8. Because some parameter configurations allow a very large particle sinking speed, which may exceed more than one box per time step, in MIT2.8 we used a biogeochemical time step length of 1/70 day for all simulations with $\eta = 1.17$ (see Table 1), in the finer resolution, ECCO1.0, we used in all

experiments a time step of 1/80 day (see Table 1) but with exception of three experiments, where we used a length of 1/160 day (these are the experiments for a strong increase of sinking speed with particle size, given by parameter $\eta = 1.17$; see Table 1). Each model was integrated for 3,000 years until tracers approached steady state. The last year is used for analysis as well as misfit calculations.

## 2.2 The biogeochemical model

### 2.2.1 Model of Oceanic Pelagic Stoichiometry

The Model of Oceanic Pelagic Stoichiometry, called MOPS (Kriest and Oschlies, 2015), is based on phosphorus, and simulates phosphate, phytoplankton, zooplankton, dissolved organic phosphorus (DOP) and detritus. The unit of each tracer is given in mmol P m$^{-3}$. In addition, MOPS simulates oxygen and nitrate. The P-cycle is coupled to oxygen by using a fixed stoichiometry of $R_{-O2:P}=171.739$, and to nitrogen by $R_{P:N}=16$.

The stoichiometry of anaerobic and aerobic remineralisation is parameterised following Paulmier et al. (2009). Remineralisation of detritus and DOM is fixed to a constant nominal remineralisation rate $r$ and is dependent on oxygen but independent of temperature. If oxygen concentrations decrease, denitrification replaces aerobic respiration, consuming nitrate. If neither oxygen nor nitrate is sufficiently available, remineralisation stops as the model does not account for other electron acceptors such as sulphate. As both forms of remineralisation follow a saturation curve (Monod-type), the realised

remineralisation rate may diverge from the constant nominal remineralisation rate.

On long timescales, the loss of fixed nitrogen through denitrification is balanced by temperature-dependent nitrogen fixation. Therefore, it should be noted that while phosphorus is conserved, the inventory of fixed nitrogen as well as oxygen is variable, and dependent on ocean circulation and biogeochemistry (Kriest and Oschlies, 2015).

In the basic model without aggregation the sinking speed of detritus increases linearly with depth. With constant

remineralisation rate $r$, the particle flux can thus be described by $F(z) \propto z^{-b}$ with $b = \frac{r}{a}$ (Kriest and Oschlies, 2008), and is therefore comparable to the common power-law description of observed particles fluxes (Martin et al., 1987). The fraction of detritus reaching the seafloor follows two pathways: One fraction is re-suspended back into the deepest box of the water column and the other one is buried into the sediment and therefore responsible for P-removal. However, the P-budget remains annually unchanged by the resupply of buried P via river runoff.



### 2.2.2 Model for particle aggregation and size dependent sinking

Different approaches have been applied to simulate particle aggregation in the marine environment. A detailed representation of the particle size spectrum can be accomplished by explicitly simulating many different size classes, which interact with each other via collision-based aggregation, particle sinking, remineralisation and breakup (Burd, 2013; Jackson, 1990). This

flexible approach captures the details of the size spectrum and its spatio-temporal variation in a very detailed way. However, it is computationally expensive, and thus prohibitive to be applied to large spatial and long temporal scales.

The aggregation module applied in MOPS parameterises a continuous log-log-linear size distribution of particles via the spectral slope $\varepsilon$ calculated from number and mass of particles (Kriest and Evans, 2000). The particle size distribution is influenced by size-dependent particle aggregation and sinking (Kriest, 2002; Kriest and Evans, 2000). Because aggregation

reduces particle numbers (but not mass), and sinking preferentially removes large particles, number and mass change independently. By assuming a log-log-linear size spectrum, the slope $\varepsilon$ of this spectrum can, at each time step and grid point, be computed from the particle number and total particle mass.

The model requires parameters for the power-law relationships between particle diameter, $d$, and mass, $m$, ($m = Cd^{\zeta}$) and between particle diameter and sinking speed, $w$, ($w = Bd^{\eta}$) to be specified. In our model experiments, we assign fixed values

for the minimum diameter and mass of a primary particle of size of $d_1$=0.002 cm and $m_1$=0.00075 nmol P. The exponent for the relationship between size and mass is set to $\zeta = 1.62$, as proposed for marine aggregates in Kriest (2002). For the relationship between size and sinking speed we test two alternative values for eta, namely $\eta = 0.62$ and $\eta = 1.17$ for the exponent, and $w_1$ between 0.7-2.8 m d$^{-1}$ for the minimum sinking speed (see below). Assuming a constant degradation rate, the average sinking speed of all particles combined would increase with depth due to higher sinking speed of large particles

and their higher proportion in the deeper ocean interior. To prevent instabilities at very large sinking speeds (very flat size distributions), as in Kriest and Evans (2000) and Kriest (2002) we restrict size dependent processes (sinking, aggregation) to a maximum size of $D_L$. In our model experiments, we let this parameter vary between 1, 2 and 4 cm.

Changes of the number of marine particles are dependent on particle aggregation, described by the collision rate, and the probability that two particles stick together, $\alpha$. In our model experiments we vary $\alpha$ between 0.2-0.8. The collision rate

depends on turbulent shear and differential sinking and is parameterised as in Kriest (2002). We assume that the turbulent shear is high in the euphotic layers and zero in the deeper ocean layers.

To avoid complications and non-linear feedbacks, in the experiments presented here, we assume that plankton mortality and zooplankton egestion as well as quadratic zooplankton mortality produce new detritus particles, but do not change the size spectrum.

By using this setup, the module is similar to parameterisations of particle size applied in other large-scale or global models (Gehlen et al., 2006; Oschlies and Kähler, 2004; Schwinger et al., 2016).





### 2.3 Model simulations and experiments

### 2.3.1 MOPS without Aggregation

As a reference scenario, we used MOPS as described in Kriest and Oschlies (2015). The model has been implemented in both global configurations MIT2.8 (hereafter called noAgg$^{MIT2.8}$) and in the finer resolution ECCO1.0 (noAgg$^{ECCO1.0}$).

### 2.3.2 Adjustment of biogeochemical model parameters

Introducing aggregates and a dynamic particle flux profile to the global model MOPS has a strong impact on biogeochemical model dynamics. Starting from parameter values of the calibrated model setup (without aggregation) of Kriest (2017), we calibrated parameters relevant for phytoplankton and zooplankton growth and turnover as described in Kriest et al. (2017) against observed global distributions of nutrients and oxygen.

Parameters to be calibrated for this new model were the light and nutrient affinities of phytoplankton, zooplankton quadratic mortality, detritus remineralisation rate, particle stickiness and the exponent $\eta$ that relates particle sinking speed to particle size (see Table 2). After introduction of particle aggregation, the calibrated nutrient affinity of phytoplankton is now much higher, with a half-saturation constant for phosphate of $K_{PHY} = 0.11$ mmol PO$_4$ m$^{-3}$ instead of 0.5 mmol PO$_4$ m$^{-3}$ in Kriest et al. (2017), very likely because the optimisation compensates for the higher export (and lower recycling) of phosphorus and

nitrogen. Possibly for the same reason, detritus remineralisation rate in the optimised model is increased from 0.05 d$^{-1}$ to 0.25 d$^{-1}$. Light affinity of phytoplankton deviates less from the value in the model without particle aggregation, but the quadratic mortality of zooplankton is strongly reduced (1.6 (mmol P m$^{-3}$)$^{-1}$ instead of 4.55 (mmol P m$^{-3}$)$^{-1}$); the latter might be regarded as an attempt of the optimisation to reduce the export of organic matter from the euphotic zone. The two parameters that affect aggregation and particle sinking remained at moderate values of $\alpha = 0.42$ and $\eta = 0.72$, i.e. close to those applied in

earlier model experiments with aggregation (e.g. Kriest, 2002). The residual cost function $J_{RMSE}$ of this pre-calibrated model with aggregation was 0.472, i.e. lower than noAgg$^{MIT2.8}$ ($J_{RMSE} = 0.529$), but somewhat higher than achieved with a model version optimised against nutrient and oxygen concentrations (Kriest, 2017), that resulted in a misfit of $J_{RMSE} = 0.439$. In the sensitivity experiment described below we will examine, whether this remaining misfit can be reduced even further, and evaluate the model sensitivity to changes in the parameters of this highly complex module.

### 2.3.3 Sensitivity experiments at coarse resolution (MIT2.8)

In the coarser model configuration of MOPS, MIT2.8, a first sensitivity study of 36 model simulations with different aggregation parameters was performed (see Table 1). We varied the values of four aggregation parameters, which control the rate of aggregation and the sinking behaviour of particles. The first parameter is the stickiness $\alpha$, i.e. the probability that after collision two particles stick together, which was set to values of 0.2, 0.5 and 0.8, respectively. The second parameter is the

maximum particle diameter for size dependent aggregation and sinking, $D_L$, set to values of 1, 2 and 4 cm. A small value of $D_L$ reduces the maximum possible sinking speed of the detrital pool, and vice versa. Parameter $w_1$ describes the sinking





speed of a primary particle with values of 0.7, 1.4 and 2.8 m d$^{-1}$. One effect of a small value of $w_l$ is that it reduces the loss of organic matter from surface layers, and thus has a direct effect on the recycling of nutrients at the surface. At the same time, it also affects the maximum possible sinking speed of the entire detritus pool. Finally, the exponent that relates particle sinking to diameter, $\eta$, is set to values of either 0.62 and 1.17. A high $\eta$ represents dense particles, and a fast increase of

particle sinking speed with size, a low value stands for more porous particles, which show only a weak relationship between size and sinking speed (Kriest, 2002).

**2.3.4 Sensitivity experiments at fine resolution (ECCO1.0)**

The occurrence of aggregates, and their transport to the ocean interior, can furthermore depend on physical dynamics (e.g. Kiko et al., 2017). Therefore, in a second step, we repeated some of the experiments presented above in the finer resolution

version ECCO1.0 to investigate possible improvements at higher resolution. In particular, we repeated all MIT2.8-simulations with $\eta$ = 0.62 in this finer resolution configuration. Additionally, we carried out three more simulations with $\eta$ = 1.17 but with the smallest $D_L$ = 1 cm to prevent particles from sinking through more than one box per time step (see Table 1). All simulations together lead to 30 model runs in the finer resolution configuration. To compare the ECCO1.0 simulations directly with results from MIT2.8, we re-gridded the result from ECCO1.0 simulations onto the coarser MIT2.8

grid.

**2.4 Model Assessment and Diagnostics**

Because observational data of particle flux are either limited with regard to space and time (e.g. Gehlen et al., 2006) or are combined with assumptions, that yield no clear patterns (Gehlen et al., 2006; Henson et al., 2012; McDonnell and Buesseler, 2010), this study restricts the model assessment to observations of nutrients and oxygen, in combination with the model fit to

volume and location of oxygen minimum zones.

**2.4.1 Root Mean Squared Error of Tracers**

After a spin-up of 3,000 years into a seasonally cycling equilibrium state, the model results are evaluated in terms of annual means of oxygen, phosphate and nitrate. As in previous studies (e.g. Kriest et al., 2017) the misfit is calculated by the deviation between simulated results, $m$, and observed properties taken from the World Ocean Atlas (WOA), $o$, (Garcia et al.,

2006). The deviations are weighted by volume of each grid box $V_i$, expressed as the fraction of the total ocean volume $V_T$. The sum of the weighted deviations is normalised by the observed global mean concentration of each tracer:

$$J_{RMSE} = \sum_{j=1}^{3} J(j) = \sum_{j=1}^{3} \frac{1}{o_j} \sqrt{\sum_{i=1}^{N} \left(m_{i,j} - o_{i,j}\right)^2 * \frac{V_i}{V_T}} \tag{1}.$$

In this equation, $j$=1,2,3 describes the respective tracer (i.e. PO$_4$, NO$_3$ and O$_2$). $N$ is the total number of model grid boxes and $o_j$ is the global average observed concentration of each tracer (Kriest et al., 2017). Thus, a low misfit value represents a good

agreement between model and observations ($J_{RMSE}$ = 0 would be a perfect fit), which enables a prediction about the model





accuracy with regard to these tracers. The model runs with the lowest $J_{RMSE}$ in the coarse and the fine resolution are called RMSE$^{MIT2.8*}$ hereafter and RMSE$^{ECCO1.0*}$, respectively.

### 2.4.2 Fit to oxygen minimum zones

To evaluate the extent and location of OMZs, we follow the approach of Cabré et al. (2015) by calculating the overlap between modelled and observed (Garcia et al., 2006; hereafter referred to as "WOA") OMZs. As several marine processes are oxygen-dependent but have heterogeneous criteria for their minimum oxygen threshold, in this study, the OMZs are calculated for different oxygen threshold concentrations, $C$. Therefore, low-oxygen waters are characterised as $O_2 < c$, with $c$ ranging from 0 to 100 mmol $O_2$ m$^{-3}$. To calculate the overlap between simulated and observed OMZs, we use the following equation (Sauerland et al., accepted):

$$C = \frac{V^{\cap}(c)}{V_U(c)} = \frac{V^{\cap}(c)}{V^m(c) + V^o(c) - V^{\cap}(c)}$$ (2).

In this equation, $V^{\cap}(c)$ is the volume of overlap of suboxic waters between model and observations, with regard to the defined oxygen threshold concentration $c$. This overlap is divided by the union (total volume of low-oxygen waters occupied in the model or in the observations) and results in a value between 0, equal to zero overlap between model and observations, and 1, which represents an optimal overlap. To adjust the scale to $J_{RMSE}$, we calculated:

$$J_{OMZ} = 1 - C$$ (3).

In this equation, $J_{OMZ}$ varies between zero and one. Consequently, the scale of $J_{OMZ}$ is equivalent to the scale of $J_{RMSE}$, which implies that a low misfit corresponds to a good agreement between model and observational data and vice versa. The model simulations with regard to lowest $J_{OMZ}$ are called OMZ$^{MIT2.8*}$ and OMZ$^{ECCO1.0*}$ hereafter. In calculating the overlap, we distinguish between the global ocean and the Pacific as well as the Atlantic Ocean.

### 2.4.3 Estimation of particle flux length scale $b$

To investigate, if, and how, the model reproduced observed maps of the particle flux length scale, $b$ derived by Marsay et al. (2015) and Guidi et al. (2015), we log-transformed the simulated, annual average flux of particulate organic matter as a function of depth and carried out a linear regression of these values. Highest $b$ values (most positive) correspond to short

particle flux length scale, i.e. many small particles, and thus a low sinking speed, shallow remineralisation and high oxygen consumption in shallow waters. For the reference models without aggregation these global maps should, in areas with shallow mixed layers, show spatially uniform values, as imposed by the model's prerequisites. Deviations from uniform values can either be ascribed to oxidant limitation of remineralisation (see above model description), or from physical processes such as mixing or upwelling, which can result in an additional vertical transport of particles.

The parameterisation of the aggregation model assumes a constant sinking speed for an upper size limit $D_L$ (see above), and therefore average particle sinking speed will remain constant below some depth. Also, the assumption of a particle size



spectrum, size dependent sinking and constant remineralisation will result in particle flux profiles that do not fully agree with those predicted by a power law (see Kriest and Oschlies, 2008). Thus, because the aggregation model's prerequisites do not fully agree with a continuous increase of sinking speed with depth, we confine the regression of log-transformed particle flux to a vertical range between 100-1000 m, where the aggregation model still shows an increase of average sinking speed with

depth (see also Kriest and Oschlies, 2008).

## 3. Results

### 3.1 Global patterns of particle flux profiles

As could be expected, noAgg$^{ECCO1.0}$ shows almost no spatial pattern of $b$, with values around the prescribed, nominal value of $b = 0.858$ (global mean: 0.64; Fig. 1a; please note the different scaling in (a) and (d)) indicating long particle flux length

scales and deep remineralisation. Regions with particularly low diagnosed $b$ values ($< 0.2$) result either from decreased remineralisation in OMZs (e.g. eastern tropical Pacific OMZ) or are found in areas of deep mixing (e.g. western boundary currents), where vertical mixing increases the inferred particle flux length scales. However, for the best simulation with regard to the sum of $J_{RMSE}$ and $J_{OMZ}$ of the aggregation model (called ECCO1.0* hereafter) we find highest (most positive) $b$ values, corresponding to short particle flux length scales, or shallow remineralisation, in the oligotrophic subtropical gyres.

In contrast, $b$ is smallest in the equatorial upwelling and in the shelf regions (close to zero; Fig. 1d and g). This pattern is in accordance with the observed spatial pattern derived by Marsay et al. (2015). In our model, this very deep flux penetration ($b$ close to zero) in the equatorial upwelling can be explained with low oxygen concentrations, which reduce the remineralisation rate. In contrast, when calibrating our model in another simulation with oxygen-independent remineralisation, we find a $b$ close to the prescribed $b$ value of 0.858 (Fig. S1).

In the subtropical and the equatorial region, the spatial variance (marked transparent red; Fig 1g) of model-derived $b$ values is quite high, which is caused by spatial variations in the physical environment, i.e. permanently stratified subtropical gyres and upwelling regions with low oxygen and reduced remineralisation. However, besides ECCO1.0* the four best model simulations with respect to the sum of $J_{RMSE}$ and $J_{OMZ}$ (simulation #14, #17, #28 and #29; Table 1) show essentially the same pattern of $b$ (Fig. S2), although these four simulations exhibit quite different parametrisations (see Table 1).

Regions with high (strongly positive) $b$ values are characterised by a high spectral slope of the size distribution and therefore a high abundance of small particles, leading to slow sinking speeds (not shown) and low export rates in ECCO1.0* (Fig. 1f). ECCO1.0* simulates highest export rates at high latitudes and in the upwelling region and lowest export rates in the subtropical gyres (Fig. 1f and i). Although the spatial pattern of export rates is similar for both model simulations with and without aggregation, ECCO1.0* shows a 1.6-fold higher global mean export rate (161 mmol P m$^{-2}$ a$^{-1}$) than noAgg$^{ECCO1.0}$ (98

mmol P m$^{-2}$ a$^{-1}$). In ECCO1.0* export rates show a higher regional variability than in noAgg$^{ECCO1.0}$ (Fig. 1c, 1f and 1i),



which is due to blooms in the high latitudes during summer season accelerating the size-dependent aggregation and thus the export signal.

The oxygen concentration at a depth of 100 m shows the same global pattern in both simulations, with high oxygen concentrations at high latitudes, and decreasing concentrations towards the equator (Fig. 1b and 1e). However, the oxygen

concentration at high latitudes is slightly higher in noAgg$^{ECCO1.0}$ than in ECCO1.0* (Fig. 1h). Moreover, the global suboxic volume (for a criterion $c = 50$ mmol m$^{-3}$) in ECCO1.0* ($7.3 \times 10^{16}$ m$^3$) is larger than in noAgg$^{ECCO1.0}$ ($3.7 \times 10^{16}$ m$^3$). Comparing our model results with the dataset of Garcia et al. (2006), which yields a volume of $5.6 \times 10^{16}$ m$^3$, we find an underestimation of the suboxic volume for noAgg$^{ECCO1.0}$ of 34% and an overestimation for ECCO1.0* of 30%.

### 3.2 Representation of oxygen minimum zones

The finer resolution and data-assimilated circulation of ECCO1.0 in general improves the representation of OMZs in comparison to MIT2.8 with regard to the overlap of OMZs for a criterion of 50 mmol m$^{-3}$ (Fig. 2). Both simulations without explicit particle dynamics, namely noAgg$^{MIT2.8}$ and noAgg$^{ECCO1.0}$, clearly underestimate the extent of the OMZ at a depth of 500 m and 1000 m for an OMZ-criterion of 50 mmol m$^{-3}$ in the Pacific basin (Fig. 2). The simulations including particle dynamics that are best with respect to the OMZ metric, OMZ$^{MIT2.8*}$ and OMZ$^{ECCO1.0*}$, exhibit a larger OMZ area for both

resolutions (Fig. 2). Despite the improved representation of OMZs, all models including the particle aggregation module still tend to merge the OMZs of the Northern Hemisphere (NH) and the Southern Hemisphere (SH) at a depth of 500 m, which does not agree with the well separated northern and southern OMZ shown by the observations (Fig. 2 and Fig. S3). As reflected in a plot that shows the extent of OMZ in the northern and southern hemisphere, similar to Fig. 1a and 1b of Cabre et al. (2015), all models fail to represent the double structure of OMZ north and south of the equator. However, in our model

the northern Pacific OMZ is fitted quite well (Fig. 2 and Fig. S3).

Aggregation improves the representation of OMZs with respect to a criterion of $c = 50$ mmol m$^{-3}$ compared to the simulations without aggregation for both resolutions in the NH, but not in the SH (Fig. 3). In noAgg$^{ECCO1.0}$ the OMZ simulated in the NH is too small and too shallow (Fig. 3a). Even though OMZ$^{ECCO1.0*}$ tends to underestimate the suboxic area between ~700 m and 1300 m, it shows a considerably higher overlap of model results and observations compared to

noAgg$^{ECCO1.0}$ (Fig. 3b). However, in the SH noAgg$^{ECCO1.0}$ represents the OMZs better than OMZ$^{ECCO1.0*}$, which tends to overestimate the suboxic area in this hemisphere. In addition to differences caused by particle dynamics, circulation affects the performance in the two hemispheres: OMZ$^{ECCO1.0*}$ represents the highest overlap between ~100 and 500 m depth in the SH but this is surpassed by OMZ$^{MIT2.8*}$ between 500 and 900 m depth. In the NH, OMZ$^{ECCO1.0*}$ outcompetes OMZ$^{MIT2.8*}$ between 300 and 900 m depth as far as overlap is concerned (Fig. 3b).

However, the improvement of the representation of OMZs in the simulations with aggregation depends on the criterion for OMZs. As could be expected, a higher oxygen threshold for the OMZ-criterion enhances the overlap between model simulations and observational data (Fig. 4). As for the fixed criterion of 50 mmol m$^{-3}$, globally and in the Pacific the better circulation and finer resolution of ECCO1.0 improves the overlap for varying OMZ-criterions in comparison to MIT2.8 (Fig.





4a and c). While the OMZ$^{\text{ECCO1.0*}}$ simulation reaches globally a maximum overlap of 65.9% (for $c$ = 100 mmol m$^{-3}$), OMZ$^{\text{MIT2.8*}}$ culminates only in a maximum of 58.7% for the same criterion.

In the Pacific basin OMZ$^{\text{ECCO1.0*}}$ reaches an agreement with observations of 19.9% overlap for a criterion of 20 mmol m$^{-3}$ (Fig. 4c). The overlap then increases strongly until the 100 mmol m$^{-3}$ criterion (68.2%). It is noteworthy that globally and in

the Pacific area noAgg$^{\text{ECCO1.0}}$ outperforms all models for a criterion of 20 mmol m$^{-3}$, where it shows an agreement of almost 31%. The Atlantic basin shows an inverse trend (Fig. 4b): here, OMZ$^{\text{MIT2.8*}}$ represents the OMZ better than OMZ$^{\text{ECCO1.0*}}$ (26% and 12.2%, respectively, for a criterion of 70 mmol m$^{-3}$). Further, in this region, the ECCO1.0 model that performs best with respect to RMSE (RMSE$^{\text{ECCO1.0*}}$) outperforms OMZ$^{\text{ECCO1.0*}}$ over the full range of criteria (Fig. 4b). Thus, there are large regional differences in the model's response to different circulations and particle dynamics. Because the dataset of

observations used for comparison does not contain any concentrations below 30 mmol m$^{-3}$ in the Atlantic, all models show no overlap at all in this basin.

In summary, the improvement of model fit with regard to $J_{OMZ}$ depends not only on particle dynamics, but also on the definition of OMZs (i.e. the OMZ criterion $c$), the model resolution as well as the region considered (Fig. 2, Fig. 3, Fig. 4).

### 3.3 Sensitivity of nutrient and oxygen distributions to aggregation parameters

Table 3 shows that in six cases out of nine (MIT2.8), a model that represents porous particles ($\eta$ = 0.62) outperforms the corresponding model with a sinking speed that describes rather dense, cell-like particles ($\eta$ = 1.17). The same applies for the higher resolution (ECCO1.0), where in two cases out of three a porous parameterisation improves the fit with regard to $J_{RMSE}$ (see Table 1). Also, both $J_{RMSE}$ and $J_{OMZ}$ of the "dense" parameterisations are never among the best five models with respect to either metric (see Table 1). Thus, in the following we focus on model simulations with $\eta$ = 0.62.

Among the sensitivity experiments performed, the best model with respect to $J_{RMSE}$ (hereafter referred to as RMSE$^{\text{MIT2.8*}}$) is characterised by an intermediate stickiness $\alpha$ of 0.5, the largest diameter for size-dependent aggregation and sinking, $D_L$, of 4 cm and a minimum particle sinking speed $w_1$ of 2.8 m d$^{-1}$, representing a rather fast organic matter transport to the ocean interior. However, many other models with medium stickiness perform about equally well (Fig. 5, upper mid panel). Models with lower stickiness perform best with slow minimum sinking speed $w_1$ and a large maximum size $D_L$=4 for size-dependent

sinking and aggregation (Fig. 5, upper left panel). In contrast, a large stickiness (which facilitates the formation of aggregates in surface layers) requires either small $w_1$ or $D_L$, which reduces the export of particles out of the euphotic zone, and into the ocean interior.

Oxygen concentrations contribute most to the global $J_{RMSE}$ (Kriest et al., 2017). The influence of oxygen on global tracer misfit is dominated by the deep concentrations (Fig. S4), and thus to a large extent by the large-scale circulation. The OMZs,

because of their small regional extent, contribute less to the global misfit (Kriest et al., 2017). This is confirmed by Fig. S4 (d, e, f), showing that, in the eastern tropical Pacific region deep (>300 m) mesopelagic and deep oxygen concentrations scatter strongly among the different models (Fig. S4 a), despite their good global match in shallow waters. Likewise, although global mean profiles of nutrients are quite similar among the different circulations, and agree quite well with





observations, their concentrations scatter strongly in the eastern tropical Pacific. Most of the simulations tend to underestimate the oxygen and nitrate concentration in this region (Fig. S4 a and c). Too low oxygen concentrations lead to too high denitrification and thus widespread nitrate depletion in the eastern tropical Pacific region, which explains the simultaneous underestimate of oxidants in this region.

To sum up, a moderate stickiness enhances the chance of a good model fit to nutrients and oxygen ($J_{RMSE}$), but there is no unique trend for the parameters or combination of parameters, with the exception of the exponent that relates particle sinking speed to its size: here, we find an advantage of a parameterisation characteristic for porous marine aggregates. In the optimal scenario, the misfit is less than that of a model without aggregates, when this is simulated with fixed reference parameters (noAgg$^{MIT2.8}$). Because of the small spatial extent of OMZs, the model fit to nutrient and oxygen concentrations is mainly

caused by the large-scale tracer distribution, even if some models show a considerable mismatch to these tracers in OMZs.

The pattern for $J_{RMSE}$ does not change very much when applying a different, higher resolved and data assimilated circulation (see Table 1 and Fig. 6). Now, the optimal model (RMSE$^{ECCO1.0*}$) is improved with respect to $J_{RMSE}$ by about 13%, but many other, almost equally good solutions, can be found with moderate to high stickiness. Introducing aggregates in this coupled model system does not improve the model fit to nutrient and tracer concentrations, as evident from the comparison of

RMSE$^{ECCO1.0*}$ ($J_{RMSE}$ = 0.431) against a model without aggregate dynamics ($J_{RMSE}$ = 0.426; Table 1). The lack of improvement can likely be explained by the fact that the biogeochemical parameters of MOPS with particle dynamics were adjusted in the circulation of MIT2.8, and thus not optimal for the model when simulated in the physical dynamics of ECCO1.0.

The sensitivity to the metric for OMZs differs from the one to the metric for nutrients and oxygen. Now, for the fit to oxygen

minimum zones ($J_{OMZ}$), a large stickiness, $\alpha$, in combination with $D_L$ of 2 cm and slow to moderate minimum sinking speed $w_I$ are of advantage (Fig. 5 and Fig. 6). Thus, a high rate of aggregation, and a maximum sinking speed of about 50-100 m d$^{-1}$ improves the model with respect to OMZs. This is also evident from comparison of the optimal models (OMZ$^{MIT2.8*}$ and OMZ$^{ECCO1.0*}$) to models without aggregate dynamics (noAgg$^{MIT2.8}$ and noAgg$^{ECCO1.0}$), shown in Fig. 3 and Fig. 4 and subsection 3.2. Nevertheless, even the models that perform best with respect to $J_{OMZ}$ underestimate mesopelagic oxygen

when averaged over the eastern tropical Pacific (Fig. S4 a).

The sensitivity patterns with regard to $J_{OMZ}$ among both configurations MIT2.8 and ECCO1.0 diverge considerably from each other, which is in contrast to the patterns for $J_{RMSE}$ noted above (compare Fig. 5 with Fig. 6). Thus, model performance with respect to $J_{OMZ}$ seems to depend much more on circulation and physical details than the large-scale dynamics reflected in $J_{RMSE}$.

**4. Discussion**

In our sensitivity study, we used a similar parameterisation of particle aggregation as Oschlies and Kähler (2004) applied in their biogeochemical-circulation model for the North Atlantic Ocean. The difference compared to our model consists in:





aggregates, which are composed of phytoplankton and detritus, the parameterisation, which is based on dense particles (dSAM, Kriest 2002) and a biogeochemical model, which is different. We found high values for the spectral slope of the size distribution (i.e. high abundance of small particles) and thus a low particle sinking speed in the subtropical gyres (Fig. S5), which corresponds with the findings by Oschlies and Kähler (2004) and Dutay et al. (2015). This, in turn, leads to highest $b$

values in the oligotrophic subtropical gyres and lowest ones (close to zero) in the high latitudes and the upwelling region, and agrees with the pattern as shown in Marsay et al. (2015). These findings imply that such a $b$ pattern cannot only result from temperature dependent remineralisation - as suggested by Marsay et al. (2015) - but also from particle dynamics and temperature-independent remineralisation. Beside particle dynamics, the low $b$ values in upwelling regions found in our study (Fig. 1d), are also caused by the suboxic conditions, which suppress remineralisation. Such a tight link between

suboxia and deep flux penetration is supported by the observations reported by Devol and Hartnett (2001) and Van Mooy et al. (2002).

However, it should be noted that the range of $b$ values in our model is larger than in most empirical studies (Berelson, 2001; Buesseler et al., 2007; Martin et al., 1987; Van Mooy et al., 2002). This is due to the fact that our model simulates too many small particles because other processes that modify the size spectrum, like the egestion of large fecal pellets by zooplankton,

are not considered, resulting in a too steep particle distribution.

As we used on the one hand two different model grid resolutions and on the other hand varied model parameterisations with regard to particle aggregation, changes in the location and extension of OMZs and the distribution of tracers within each resolution are exclusively driven by the aggregation parameters. A good parameterisation of particle aggregation parameters can therefore have a major influence on the representation of OMZs. Furthermore, a higher model resolution improves the

depiction of equatorial currents and therefore the oxygen transport (Cabré et al., 2015; Duteil et al., 2014), which, in turn, results in an improved representation of OMZs in the finer resolution configuration, ECCO1.0, compared to the coarser resolution, MIT2.8. However, as physical processes at smaller scales affect the simulated shallow to mesopelagic oxygen and nutrient concentrations for the eastern tropical Pacific (Getzlaff and Dietze, 2013), the finer (1°x1°) resolution of ECCO1.0 is not sufficient to resolve the details of the equatorial current system (Duteil et al., 2014). This can explain the

still high residual misfit of these simulations, and the missing double structure of OMZs in the Eastern Tropical Pacific.

Furthermore, results of our sensitivity study confirm that dense particles do not constitute a realistic representation of particles, as indicated by Karakaş et al. (2009) and Kriest (2002). Porous particles seem to constitute a more appropriate parameterisation for good model fit with regard $J_{RMSE}$ and $J_{OMZ}$ (Table 1). Although the observed stickiness ranges between almost zero and one (e.g. Alldredge and McGillivary, 1991; Kiørboe et al., 1990), in our study a moderate stickiness,

$\alpha$, between 0.5 and 0.8 leads the model towards a good fit to observed nutrients, oxygen and OMZs.

In summary, our study supports the results of Schwinger et al. (2016), who found an improved representation of nutrient distribution and OMZs when switching from constant particle sinking to either a power law or particle dynamics, similar to those presented here. However, the difference between the two latter schemes in that study were only small. A more extensive search of the parameter space within a given circulation may have further improved that model. Additionally, we



optimised noAgg[MIT2.8] against the same misfit function as MOPS[oD] of Kriest et al. 2017 and found that even though including an aggregation module improves our model, utilising an appropriate parameter optimisation would further enhance our model fit. Thus, without a comprehensive calibration of biogeochemical and aggregation parameters there only seems to be a slight advantage when using this more complex model of particle dynamics.

Finally, we found a steep particle size spectrum in the subtropical oligotrophic region (Fig. 1d), which does not agree with observational data. Potentially, there are processes taking place, which are not considered in our model i.e. particle repackaging and active transport by zooplankton (vertical migration) (Kiko et al. 2017) based on a modified food web. Thus, particle aggregation seems not to be sufficient for a correct representation of the particle size spectrum.

## 5. Conclusion and Outlook

Najjar et al. (2007) applied different model circulations to the same biogeochemical model, and found that that physical processes are an important factor for modelling marine biogeochemistry. Our study furthermore showed that also biogeochemical parameterisations - in particular, those related to particle flux - can have an important impact on the representation of dissolved inorganic tracers, in line with earlier studies (e.g. Kriest et al., 2012; Kwon and Primeau, 2006, 2008). These earlier studies applied and varied a globally uniform particle flux length scale, whereas it has been suggested
that this parameter should vary in space and time (e.g. Guidi et al., 2015; Marsay et al., 2015). The sensitivity study presented here constitutes a first approach to systematically estimate the impact of marine particle aggregation - and thus a spatially and temporally variable flux length scale - on the location and extent of OMZs as well as the representation of phosphate, nitrate and oxygen under steady-state conditions in a global three-dimensional biogeochemical ocean model. We have shown that the assumptions inherent in the model confirm the general pattern of the spatial map of $b$ values
proposed by Marsay et al. (2015) (Fig. 1a and d). This, in turn, shows that the pattern of Martin's $b$ cannot only be depicted by a POC flux dependent on temperature but also by simulating explicit particle dynamics.

We furthermore found that even though there are still a lot of gaps in understanding several processes e.g. the variation of export rates, particle stickiness and particle flux profile over space and time, as well as the link between particle diameter and sinking speed, the comparisons against observational data show a trend towards a model improvement by integrating
particle dynamics (Table 1). While the parameterisation of aggregation leads the model towards an improved fit to OMZs for both model resolutions, this increase in model fit with regard to phosphate, nitrate and oxygen is only detectable in the coarse resolution MIT2.8, but not in the finer resolution and data-assimilated circulation of ECCO1.0. Moreover, model simulations show that besides effects of grid resolution, the model fit with regard to $J_{RMSE}$ and $J_{OMZ}$ is mainly driven by the particles' porosity. Our results indicate that a best fit to both, tracers as well as OMZs (50 mmol $O_2$ m$^{-3}$ criterion), is
achieved by parameterising porous particles in combination with an intermediate to large maximum particle diameter for size dependent aggregation and sinking, a moderate to high stickiness ranging between 0.5 and 0.8 and an intermediate to high initial sinking speed ranging between 1.4 and 2.8 m d$^{-1}$ (Fig. 5). The strong sensitivity of the model fit to aggregation



parameters may point towards the importance of a spatially and temporally varying flux length scale; however, they also show, that the dynamics of the model depend strongly on the assumptions we make with respect to particle properties and processes.

Finally, we have shown that uncertainties in the parameterisation of particle aggregation remain, leading to the inference that
dissolved inorganic tracers offer only insufficient observational constraints for global particle parameterisation. Therefore, for an accurate representation it will be necessary to calibrate the model not only against observed phosphate, nitrate, oxygen distributions and volume and location of OMZs (Sauerland et al., accepted) but also against number and size of particles, using comprehensive datasets of observations (as in Guidi et al., 2015).

**Code and data availability**

The source code of MOPS including the aggregation module coupled to TMM as well as the model output are available at: https://data.geomar.de/thredds/catalog/open_access/niemeyer_et_al_2019_bg/catalog.html.

**Acknowledgments**

This work is a contribution to DFG-supported project SFB 754 (www.sfb754.de). Parallel supercomputing resources have
been provided by the North-German Supercomputing Alliance (HLRN) and the computing centre at Kiel University.

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



**Tables**

**Table 1: Model runs of sensitivity study, their parameter combinations and the calculated misfit of tracers ($J_{RMSE}$) and OMZs ($J_{OMZ}$) for MIT2.8 and the ECCO1.0 configurations. The best simulations with regard to $RMSE^{MIT2.8}$, $RMSE^{ECCO1.0}$, $OMZ^{MIT2.8}$ and $OMZ^{ECCO1.0}$ are highlighted in red ($J_{RMSE}$) and blue ($J_{OMZ}$). OMZ is defined as 50 mmol m$^{-3}$.**

| Run | $\eta$ | $\alpha$ | $w_1$ | $D_L$ | $w_{max}$ | $J_{RMSE}$ MIT2.8 | $J_{RMSE}$ ECCO1.0 | $J_{OMZ}$ MIT2.8 | $J_{OMZ}$ ECCO1.0 |
|---|---|---|---|---|---|---|---|---|---|
| 1 | 0.62 | 0.2 | 0.7 | 1.0 | 33 | 0.894 | 0.631 | 0.817 | 0.754 |
| 2 | 0.62 | 0.2 | 0.7 | 2.0 | 51 | 0.730 | 0.499 | 0.791 | 0.739 |
| 3 | 0.62 | 0.2 | 0.7 | 4.0 | 78 | 0.531 | 0.440 | 0.817 | 0.710 |
| 4 | 0.62 | 0.2 | 1.4 | 1.0 | 66 | 0.938 | 0.735 | 0.836 | 0.767 |
| 5 | 0.62 | 0.2 | 1.4 | 2.0 | 101 | 0.823 | 0.640 | 0.805 | 0.748 |
| 6 | 0.62 | 0.2 | 1.4 | 4.0 | 156 | 0.655 | 0.535 | 0.791 | 0.736 |
| 7 | 0.62 | 0.2 | 2.8 | 1.0 | 132 | 1.033 | 0.879 | 0.919 | 0.844 |
| 8 | 0.62 | 0.2 | 2.8 | 2.0 | 203 | 1.032 | 0.877 | 0.919 | 0.844 |
| 9 | 0.62 | 0.2 | 2.8 | 4.0 | 312 | 1.030 | 0.874 | 0.817 | 0.845 |
| 13 | 0.62 | 0.5 | 0.7 | 1.0 | 33 | 0.714 | 0.510 | 0.771 | 0.737 |
| 14 | 0.62 | 0.5 | 0.7 | 2.0 | 51 | 0.561 | 0.441 | 0.730 | 0.601 |
| 15 | 0.62 | 0.5 | 0.7 | 4.0 | 78 | 0.618 | 0.567 | 0.919 | 0.585 |
| 16 | 0.62 | 0.5 | 1.4 | 1.0 | 66 | 0.603 | 0.457 | 0.778 | 0.721 |
| 17* | 0.62 | 0.5 | 1.4 | 2.0 | 101 | 0.508 | 0.443 | 0.759 | 0.580 |
| 18 | 0.62 | 0.5 | 1.4 | 4.0 | 156 | 0.848 | 0.627 | 0.919 | 0.652 |
| 19 | 0.62 | 0.5 | 2.8 | 1.0 | 132 | 0.775 | 0.693 | 0.828 | 0.760 |
| 20 | 0.62 | 0.5 | 2.8 | 2.0 | 203 | 0.570 | 0.566 | 0.805 | 0.745 |
| 21 | 0.62 | 0.5 | 2.8 | 4.0 | 312 | 0.493 | 0.459 | 0.817 | 0.737 |
| 25 | 0.62 | 0.8 | 0.7 | 1.0 | 33 | 0.690 | 0.495 | 0.748 | 0.719 |
| 26* | 0.62 | 0.8 | 0.7 | 2.0 | 51 | 0.570 | 0.465 | 0.723 | 0.551 |





| 27 | 0.62 | 0.8 | 0.7 | 4.0 | 78 | 0.667 | 0.622 | 0.936 | 0.644 |
|---|---|---|---|---|---|---|---|---|---|
| 28 | 0.62 | 0.8 | 1.4 | 1.0 | 66 | 0.522 | 0.431 | 0.758 | 0.605 |
| 29 | 0.62 | 0.8 | 1.4 | 2.0 | 101 | 0.661 | 0.560 | 0.620 | 0.578 |
| 30 | 0.62 | 0.8 | 1.4 | 4.0 | 156 | 1.011 | 0.791 | 0.936 | 0.814 |
| 31 | 0.62 | 0.8 | 2.8 | 1.0 | 132 | 0.501 | 0.456 | 0.788 | 0.727 |
| 32 | 0.62 | 0.8 | 2.8 | 2.0 | 203 | 0.682 | 0.443 | 0.708 | 0.728 |
| 33 | 0.62 | 0.8 | 2.8 | 4.0 | 312 | 1.004 | 0.597 | 0.805 | 0.656 |
| 10 | 1.17 | 0.2 | 0.7 | 1.0 | 1007 | 0.780 | 0.654 | 0.927 | 0.868 |
| 11 | 1.17 | 0.2 | 1.4 | 1.0 | 2013 | 0.945 | | 0.936 | |
| 12 | 1.17 | 0.2 | 2.8 | 1.0 | 4027 | 1.028 | | 0.919 | |
| 22 | 1.17 | 0.5 | 0.7 | 1.0 | 1007 | 0.606 | 0.506 | 0.932 | 0.927 |
| 23 | 1.17 | 0.5 | 1.4 | 1.0 | 2013 | 0.677 | | 0.912 | |
| 24 | 1.17 | 0.5 | 2.8 | 1.0 | 4027 | 0.930 | | 0.911 | |
| 34 | 1.17 | 0.8 | 0.7 | 1.0 | 1007 | 0.698 | 0.521 | 0.947 | 0.949 |
| 35 | 1.17 | 0.8 | 1.4 | 1.0 | 2013 | 0.595 | | 0.894 | |
| 36 | 1.17 | 0.8 | 2.8 | 1.0 | 4027 | 0.784 | | 0.895 | |
| noAgg | | | | | 234 | 0.529 | 0.426 | 0.791 | 0.640 |





**Table 2: Model adjustment of biogeochemistry with aggregates compared to Kriest et al. (2017) and new parameters in this study.**

| Parameters that remain fixed | Kriest et al. (2017) | this study | unit | description |
|---|---|---|---|---|
| ro2ut | 171.7 | 171.7 | mol $O_2$:mol P | Redfield ratio |
| Subdin | 15.8 | 15.8 | mmol $NO_3$ m$^{-3}$ | no denitrification below this level |
| Nfix | 1.19 | 1.19 | μmol N m$^{-3}$ d$^{-1}$ | N fixation |
| ACkbaco2 | 1.00 | 1.00 | mmol $O_2$ m$^{-3}$ | half. sat.-constant for oxic degradation |
| ACkbacdin | 31.97 | 31.97 | mmol $NO_3$ m$^{-3}$ | half sat.-constant for suboxic degradation |
| ACmuzoo | 1.89 | 1.89 | 1 d$^{-1}$ | max. grazing rate |
| **Parameters that changed compared to Kriest et al. (2017)** | | | | |
| ACik | 9.65 | 6.52 | W m$^{-2}$ | light half-saturation constant |
| ACkpo4 | 0.5 | 0.106 | mmol P m$^{-3}$ | half-saturation constant for $PO_4$ uptake |
| AComniz | 4.55 | 1.6 | m$^3$ (mmol P * day)$^{-1}$ | quadratic zooplankton mortality |
| detlambda | 0.05 | 0.25 | 1 d$^{-1}$ | detritus remineralisation rate |
| **New parameters for the aggregation model (further modified in this study)** | | | | |
| SinkExp | - | 0.7164 | | exponent that relates particle sinking speed to diameter |
| Stick | - | 0.4162 | | stickiness for interparticle collisions |




**Table 3: Number of simulations with different parameters for $D_L$, $\alpha$ and $w_I$ for the porous ($\eta = 0.62$) and dense ($\eta = 1.17$) particles which outperform the corresponding other size. The numbers are given with respect to two different criteria, $J_{RMSE}$ and $J_{OMZ}$.**

|  | $\eta = 0.62$ | $\eta = 1.17$ | resolution |
|---|---|---|---|
| $J_{RMSE}$ | 6 | 3 | MIT2.8 |
| $J_{OMZ}$ | 8 | 1 | MIT2.8 |
| $J_{RMSE}$ | 2 | 1 | ECCO1.0 |
| $J_{OMZ}$ | 2 | 0 | ECCO1.0 |





Figures

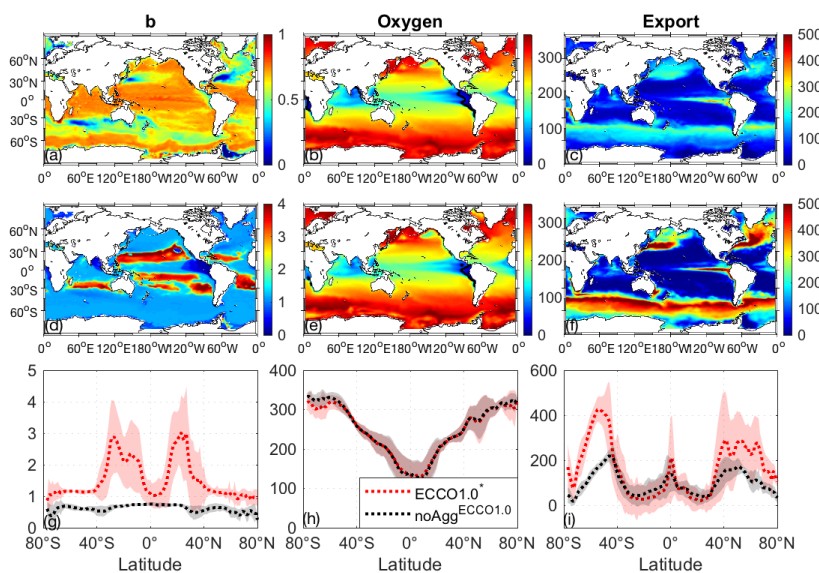

**Figure 1: Global maps of b (left panels (a) and (d)), O$_2$ at 100 m (mmol m$^{-2}$, middle panels (b) and (e)) and export at 100 m (mmol P m$^{-2}$ a$^{-1}$, right panels (c) and (f)) for noAgg$^{ECCO1.0}$ (top panels (a), (b) and (c)) and for the best aggregation model with regard to the sum of $J_{RMSE*}$ and $J_{OMZ*}$ (simulation #26; panels (d), (e) and (f)). The black line indicates the OMZ for a criterion of 50 mmol m$^{-3}$. Lower panels: Global mean (dotted line) and standard deviation (transparent shaded) of b (panel g), O$_2$ (panel h)) and export (panel i)) of noAgg$^{ECCO1.0}$ (black) and the best aggregation model with regard to the sum of $J_{RMSE*}$ and $J_{OMZ*}$ (simulation #26; red). Please note the different scaling for b-values ((a) and (d)).**



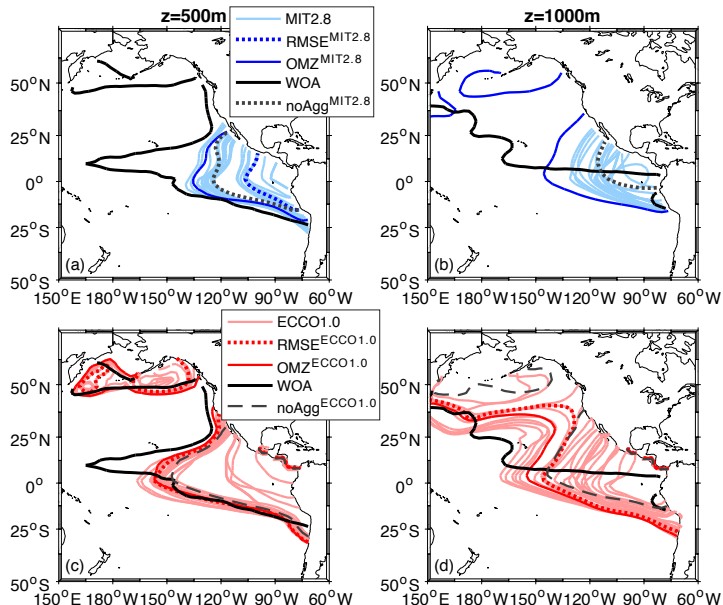

**Figure 2:** Comparison of Pacific Ocean OMZ (O$_2$ <= 50 mmol m$^{-3}$) between model simulations and observations. Panels (a) and (b) show the OMZ at a depth of 500 m and 1000 m for the coarse resolution, MIT2.8, and panels (c) and (d) for the fine resolution, ECCO1.0.



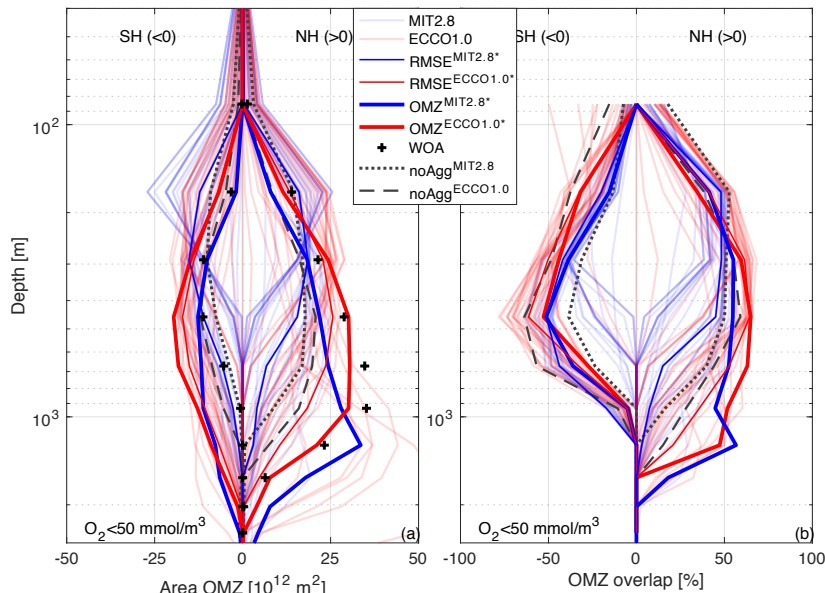

**Figure 3: Area of OMZ (left panel) and overlap of OMZs between model and observations following Cabre et al. (2015) (right panel). In both panels, the left-hand side shows the Southern Hemisphere (0-40 °S), the right shows the Northern Hemisphere (0-40 °N), plotted against the logarithmic depth. OMZs are defined as regions with $O_2<50$ mmol m$^{-3}$.**





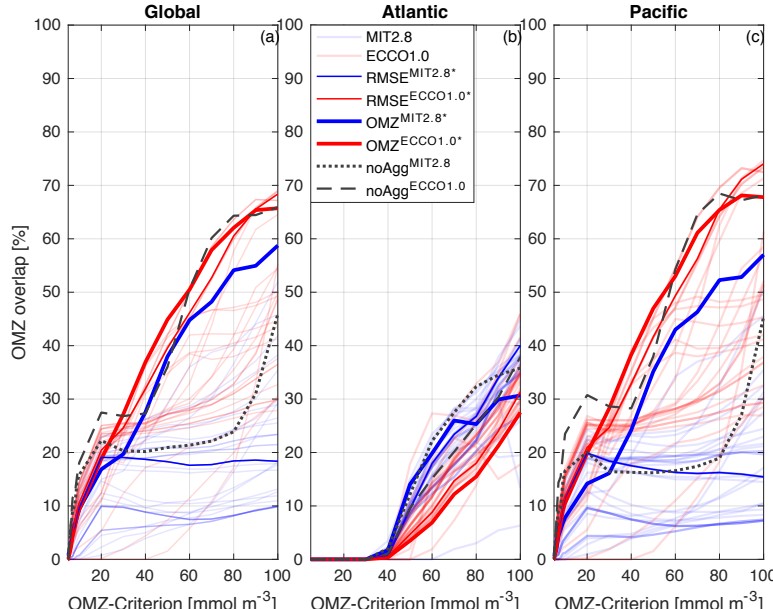

**Figure 4: Overlap between modelled and observed OMZs (Eq. (2)) for varying criteria c, ranging from $0 < c < 100$ mmol m$^{-3}$) on a global scale (left panel), for the Atlantic Ocean (middle panel) and for the Pacific Ocean (right panel).**




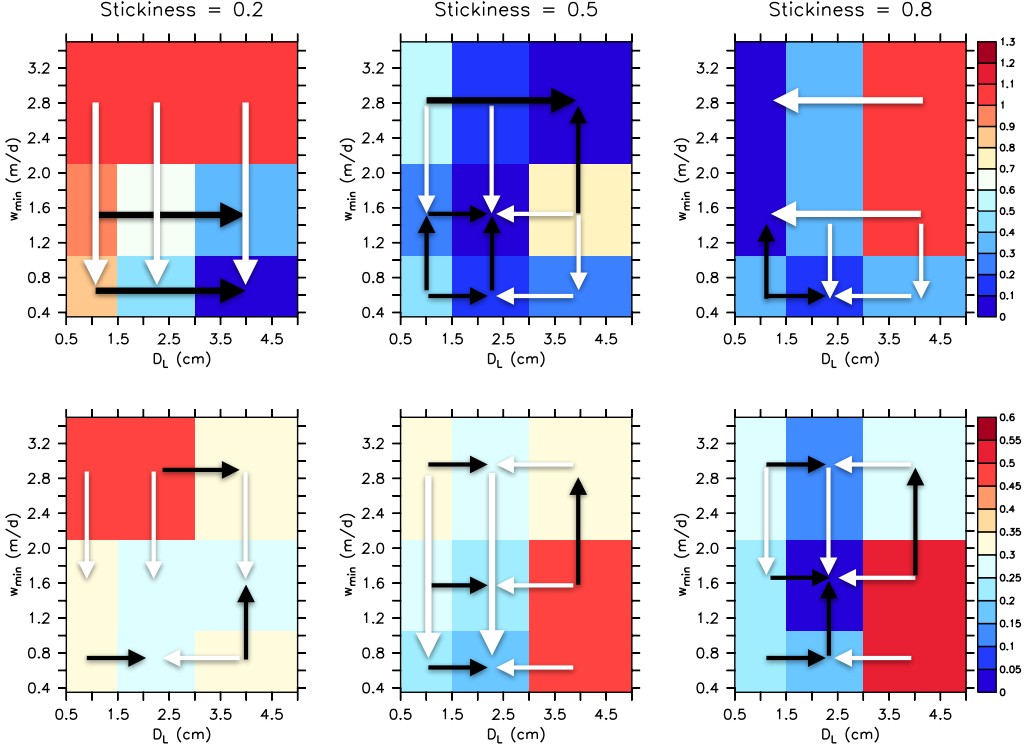

**Figure 5: Sensitivity of $J_{RMSE}$ (Eq. (1)); upper panels) and $J_{OMZ}$ (Eq. (3)); lower panels) to minimum sinking speed $w_1$ and maximum size $D_L$ for the coarse resolution MIT2.8, for three different values of stickiness (left to right), and $\eta = 0.62$ ("porous" particles). The colorbar shows $J_{RMSE}$ and $J_{OMZ}$ (blue = good fit, red = bad fit), normalised by its minimum value across all model experiments. Black arrows indicate an improvement of $J_{RMSE}$ or $J_{OMZ}$ with increasing parameter values, while white arrows show an improvement with decreasing values.**



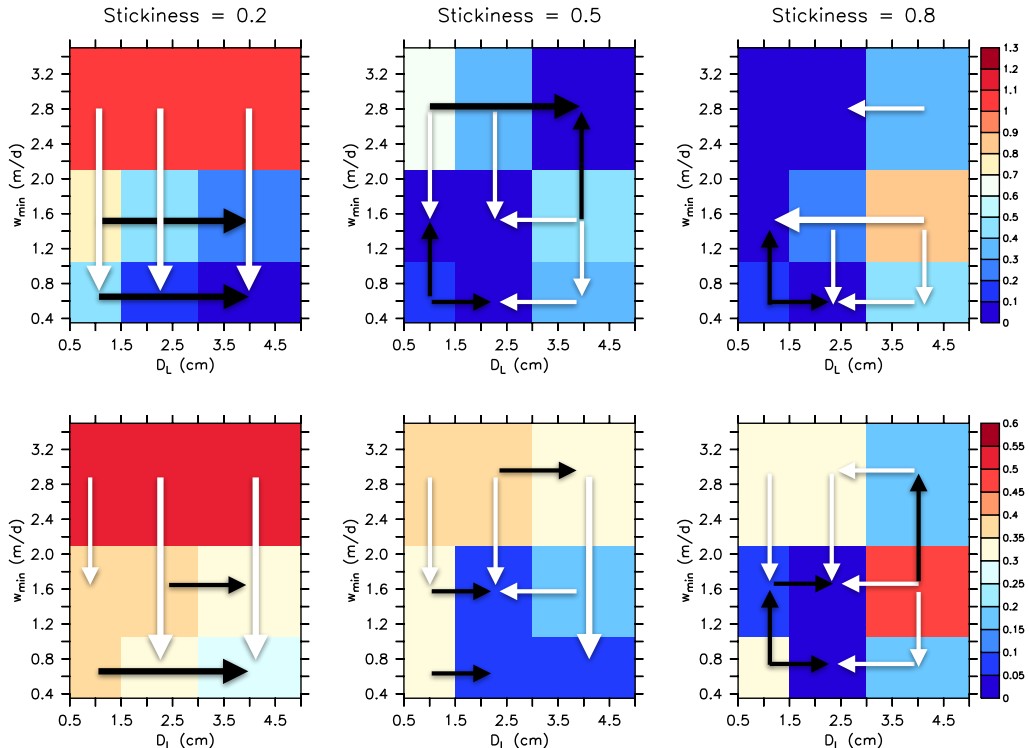

**Figure 6: As Fig. 5, but for simulations with ECCO1.0.**