# Peer review of "The effect of marine aggregate parameterisations on nutrients and oxygen minimum zones in a global biogeochemical model"

_Biogeosciences, 2019_

## Referee Comment (RC1) · Anonymous Referee #1 · 29 Apr 2019

This is a nicely focussed study looking at the parameterization of particle aggregation processes affect ocean biogeochemistry and the characteristics of oxygen minimum zones. There has been an understanding for quite a while that the way in which particle processes are represented in biogeochemical models can affect OMZs (see e.g. Moore et al., J. Climate 26:9291–9312, 2013), so it is nice to see this being addressed directly.

Whilst I enjoyed the manuscript a great deal, I did find the presentation confusing in a few places (mentioned below) and the authors should consider small re-writes to explain more explicitly what is going on.

I have some questions concerning the methodologies used but the authors. Firstly, the

mineralization of detritus is set in the model to be dependent on oxygen availability, but not temperature. Iversen and Ploug (Biogeosciences 10:4073–4085, 2013) show that temperature can have a strong influence on carbon specific respiration rates and consequently on deep-ocean particle fluxes. I can understand not including this in the model, but I think some discussion of how this might affect the results of the simulations should be included.

I find it curious that including aggregation improves representation of the OMZs using a criterion of 50 mol m-3 for both resolutions of model in the northern hemisphere, but not the southern hemisphere. I almost get the the impression that the authors are arguing that there is a fundamental difference in aggregation between the two hemispheres. If this is so, then they need to explain more explicitly what they mean. I could imagine differences in particle production, or community composition, leading to different particle processes or rates, but I wonder if this is what was meant.

The authors find that the relationship between sinking velocity and particle size is crucial, and that better fits of their model are obtained using more formulations with more porous particles. Their best fit is with eta=0.62 and using the formulae in Stemmann et al., 2004 for settling velocity as a function of mass and fractal dimension, this implies a fractal dimension of 1.62. This is in line with recent observations by Jackson who shows that his aggregation models give a best fit data for fractal dimensions of about 1.8.

The best fit to JRMSE has a maximum particle size of 4 cm, which is quite large. Is there observational evidence that such particles are common? Given the slope of the size spectrum in their models, can the authors calculate how common these particles are and where they would be found, and is there observational evidence for this?

---

## Referee Comment (RC2) · Anonymous Referee #2 · 3 Jun 2019

The authors present a sensitivity study of the impact of particle aggregation on the global performance of a biogeochemical model, with a large focus on the improvement of the representation of the OMZs. While I enjoyed the reading and I think that the model description and sensitivity analysis is a significant step forward in the field, I have several general comments that should be addressed before publications. Title: The title is too broad and I think that the mentioned to OMZs should appears somewhere because most of the sensitivity analysis is directed toward the improvement of their representation (even though O2 is not the only tracer considered)

Particle sinking speed: The introduction refers to a large range of particles sinking

speed as a function of size (which is true) spanning from 10 to 386m.d-1 (or more). I would have liked to see in the paper (at least in the results or discussion) how the model sinking speed scales with actual data (or sinking speed from other models). For example, page 5, line 18, the minimum sinking speed is mentioned to be between 7 to 2.8m.d-1 for particles of ∼0.002cm but what are the maximum values? Figure S5 shows a latitudinal section of mean sinking speed of detritus at both 100 and 500m. Values range between 0 to 600m.d-1 and 350 to 850m.d-1 at 100 and 500m respectively which are relatively high compared to whatever has been measured and published in the literature (ex. Jouandet et al., 2011, figure 8). Having this discussed in the paper would be a plus for the validation of the model.

Particle length scale b: The author acknowledges that b values in their models (the one including aggregation) is much larger than in most empirical studies (page 13 L13-15). This is an important limitation of the model and its ability to represent the extension of the OMZs. I would have like to see a better discussion on this limitation and in particular how far their bs are from empirical observations. For example, values from Marsay et al., 2015 and Guidi et al., 2015 both range between 0 and 2 even though showing different patterns. The current study present bs ranging from ∼1 up to 4 (Figure 1g) with different amplitude and absolute values and therefore important implication for both the horizontal and vertical representation of the OMZs. One explanation from the author is that the model generates too many small particles because processes such as repackaging, egestion and others are not represented. This could be true but the model also generate large aggregates (up to 4 cm) and sinking average sinking speed (see above) are fairly high. So, these 2 results are inconsistent and I would have like to see a mode developed discussion about this in the article.

Particle size distribution (slope): There is no comparison of the model size spectrum (slope at least) to actual in-situ measurements of particle size distribution which are increasingly available in the literature (ex. Kiko et al., 2017). I would have like to see this comparison in the paper to present evidence that the dynamic of particle aggregation

is well capture by the model before to perform any sensitivity analysis.

Specific comments

Page 3 L 15 and L 20: Are you referring to Marsay et al., 2015 and Guidi et al., 2015 or Henson et al., 2015 and Marsay et al., 2015 as stated?

Table 1 is very hard to go through even though very informative. Representing the 4-last column with a clustergram (heatmap) could help to cluster simulations that present similar outcomes.

---

## Author Response (AR1)

*Interactive comment on*

**The effect of marine aggregate parameterisations on nutrients and oxygen minimum zones in a global biogeochemical model**

Daniela Niemeyer[1], Iris Kriest[1], Andreas Oschlies[1]

[1]GEOMAR Helmholtz-Zentrum für Ozeanforschung Kiel, Düsternbrooker Weg 20, 24105 Kiel, Germany

*Correspondence to: Daniela Niemeyer (dniemeyer@geomar.de)*

**Responses to Referee#1**

First of all, we would like to thank the reviewer very much for his/her thoughtful and constructive comments. Below is our detailed reply. (Referee's comments in blue.)

This is a nicely focussed study looking at the parameterization of particle aggregation processes affect ocean
biogeochemistry and the characteristics of oxygen minimum zones. There has been an understanding for quite a while that the way in which particle processes are represented in biogeochemical models can affect OMZs (see e.g. Moore et al., J. Climate 26:9291–9312, 2013), so it is nice to see this being addressed directly.
Whilst I enjoyed the manuscript a great deal, I did find the presentation confusing in a few places (mentioned below) and the authors should consider small re-writes to explain more explicitly what is going on.
I have some questions concerning the methodologies used but the authors.

Firstly, the mineralization of detritus is set in the model to be dependent on oxygen availability, but not temperature. Iversen and Ploug (Biogeosciences 10:4073–4085, 2013) show that temperature can have a strong influence on carbon specific respiration rates and consequently on deep-ocean particle fluxes. I can understand not including this in the model, but I think
some discussion of how this might affect the results of the simulations should be included.
Thank you very much for this comment. We included only oxygen-dependent remineralisation and neglected the temperature-dependent remineralisation in an attempt to isolate the potential effect of particle aggregation on remineralisation and particle distribution. If temperature-dependent remineralisation was included, this would additionally impact the depth profile of particle fluxes, and a separation of both effects would be made difficult. Both, temperature-
dependent remineralisation and particle aggregation suggest deep particle flux in high latitudes as inferred from observations (e.g. Marsay et al., 2015). By excluding direct temperature effects on remineralisation, we can test the hypothesis to what extent observed regional variations in $b$ could be generated by aggregation rather than by temperature as assumed previously. In our revised manuscript, we integrated this aspect into the discussion.

"However, if temperature-dependent remineralisation, as suggested by Marsay et al. (2015) or Iversen and Ploug (2013), was also included in our model, this would likely enhance horizontal variations in the particle flux profile, with even deeper flux penetration in the cold waters of the high latitudes and upwelling areas. […] Therefore, two different processes – particle aggregation and/or temperature-dependent remineralisation – suggest low $b$ values and deep flux penetration in the very productive areas of high latitudes. A third process, which consists in oxygen-dependent remineralisation, is superimposed on
these in OMZs, causing the steepest particle profiles in these areas."

I find it curious that including aggregation improves representation of the OMZs using a criterion of 50 mmol m-3 for both resolutions of model in the northern hemisphere, but not the southern hemisphere. I almost get the impression that the authors are arguing that there is a fundamental difference in aggregation between the two hemispheres. If this is so, then they need to explain more explicitly what they mean. I could imagine differences in particle production, or community composition, leading to different particle processes or rates, but I wonder if this is what was meant.

Thank you. As already described in Cabre at al. (2015), there are several physical causes, which might explain OMZ biases.

One possible factor for this bias consists in a too weak equatorial current system affecting nutrient trapping (Dietze & Loeptin, 2013), a positive feedback between high productivity in upwelling areas and thus strong sinking and remineralisation (Schmittner et al., 2005). Together with the meridional overturning circulation, which is asymmetric with respect to the equator, this could affect the size and location of OMZs, which we suggest to be the case in our study. As we used the Transport Matrix Method (TMM), an efficient offline method that simulates total tracer transport but does not allow its separation into the different components of advection and diffusion, we are unfortunately not able to prove our suggestion. We therefore think that the difference in the representation of OMZs between NH and SH is more caused by physics than by biology. However, we decided to plot the overlap between model and observations, following Cabre at al. (2015), to be able to compare our model results with the results from CMIP models.

The authors find that the relationship between sinking velocity and particle size is crucial, and that better fits of their model are obtained using more formulations with more porous particles. Their best fit is with eta=0.62 and using the formulae in Stemmann et al., 2004 for settling velocity as a function of mass and fractal dimension, this implies a fractal dimension of 1.62. This is in line with recent observations by Jackson who shows that his aggregation models give a best fit data for fractal dimensions of about 1.8.

Thank you for your advice. In the revised manuscript, we now also refer to G. A. Jackson's work.

"The exponent for the relationship between size and mass is set to $\zeta = 1.62$, as proposed for marine aggregates in Kriest (2002), which is in line with more recent findings (Burd and Jackson, 2009; Jouandet et al., 2014)."

The best fit to JRMSE has a maximum particle size of 4 cm, which is quite large. Is there observational evidence that such particles are common? Given the slope of the size spectrum in their models, can the authors calculate how common these particles are and where they would be found, and is there observational evidence for this?

Thank you for your important comment. $D_L$ is defined as the maximum diameter for size dependent processes and aggregation. In practice, $D_L$ defines the upper limit for these processes i.e. for particles with diameter > 4 cm the sinking speed will not increase anymore. We defined this parameter more clearly in our methods and discussion.

Moreover, we find lowest values for the spectral slope in the Southern Ocean and, in line with these results, the highest abundance of large particles with a diameter > 4 cm in the Southern Ocean. Globally, we find a maximum number of these large particles of 0.0016 particles per litre. This value seems negligible small, which, in turn, is in line with observations, where the probability of collecting such large particles is quite low.

"However, model calibration against observed particle dynamics has to account for characteristics and limitations of observations. For example, the size spectrum assumed in our model is of infinite upper size and also contains particles with a diameter larger than, e.g. 4 cm (the upper limit for size-dependency of aggregation and sinking). While these particles exist (e.g. Bochdansky and Herndl, 1992), they are very rare (in the model, and likely also in the observations), and might not be observed with standard methods, which usually rely on a sample size of few litres. The rare occurrence of large particles, and the limited sample size has, for example, consequences for estimated of size spectra parameters (Blanco et al., 1994). Thus, any model calibration against observations of particle abundance and size has to account for a proper match between simulated and observed quantities."

**Responses to Referee#2**

First of all, we would like to thank the reviewer very much for his/her thoughtful and constructive comments. Below is our detailed reply (Referee's comments in blue.)

The authors present a sensitivity study of the impact of particle aggregation on the global performance of a biogeochemical model, with a large focus on the improvement of the representation of the OMZs. While I enjoyed the reading and I think that the model description and sensitivity analysis is a significant step forward in the field, I have several general comments that should be addressed before publications.

Title: The title is too broad and I think that the mentioned to OMZs should appears somewhere because most of the sensitivity analysis is directed toward the improvement of their representation (even though O2 is not the only tracer considered).

Thank you for this advice. We entitled our revised manuscript 'The effect of marine aggregate parameterisations on nutrients and oxygen minimum zones in a global biogeochemical model'.

Particle sinking speed: The introduction refers to a large range of particles sinking speed as a function of size (which is true) spanning from 10 to 386m.d-1 (or more). I would have liked to see in the paper (at least in the results or discussion) how the model sinking speed scales with actual data (or sinking speed from other models). For example, page 5, line 18, the minimum sinking speed is mentioned to be between 7 to 2.8m.d-1 for particles of ~ 0.002cm but what are the maximum values? Figure S5 shows a latitudinal section of mean sinking speed of detritus at both 100 and 500m. Values range between 0 to 600m.d-1 and 350 to 850m.d-1 at 100 and 500m respectively which are relatively high compared to whatever has been measured and published in the literature (ex. Jouandet et al., 2011, figure 8). Having this discussed in the paper would be a plus for the validation of the model.

Thank you for this important comment. Please find in Table 1 column 6 the maximum possible sinking speeds for each simulation, which is defined by $w_L = w_1 * (d_L/d_1)^\eta$. Although the maximum prescribed sinking speed depends on parameterisation and has a broad range between 33 m d$^{-1}$ (porous particles) and 4027 m d$^{-1}$ (dense particles), our best simulations with regard to $J_{RMSE}$ and $J_{OMZ}$ range between 101 m d$^{-1}$ (#17) and 51 m d$^{-1}$ (#26), which is in line with the findings by Alldredge and Gotschalk (1988), Nowald et al. (2009) and Jouandet et al. (2011). Unfortunately, we made a mistake in converting our fluxes in Figure S5, which is now Figure 7 in the revised manuscript, resulting in diagnosed sinking speeds, which were too high by a factor of approximately 16. Figure 7 now shows the correct diagnosed sinking speeds (note that the scale of the y-axis has changed). This leads to a maximum sinking speed of 50.7 m d$^{-1}$ at 500 m depth, which now agrees with column 6 in Table 1 and previous observations. Because of this mistake in converting diagnosed fluxes and properties, we also had to adapt Fig. 1c, f and i showing the export rates. These are now also lower by a factor of 16. However, in both cases the global pattern remains.

At this point, we would also like to note that the sinking speeds shown in Fig 7 are the average speed across the entire size spectrum, i.e. not directly comparable to sinking speeds of individual particles or aggregates.

Particle length scale b: The author acknowledges that b values in their models (the one including aggregation) is much larger than in most empirical studies (page 13 L13-15). This is an important limitation of the model and its ability to represent the extension of the OMZs. I would have like to see a better discussion on this limitation and in particular how far their bs are from empirical observations. For example, values from Marsay et al., 2015 and Guidi et al., 2015 both range between 0 and 2 even though showing different patterns. The current study present bs ranging from ~ 1 up to 4 (Figure 1g) with different amplitude and absolute values and therefore important implication for both the horizontal and vertical representation of the OMZs. One explanation from the author is that the model generates too many small particles because processes such as repackaging, egestion and others are not represented. This could be true but the model also generate large aggregates (up to 4 cm) and sinking average sinking speed (see above) are fairly high. So, these 2 results are inconsistent and I would have like to see a mode developed discussion about this in the article.

Thank you for your advice. $D_L$ is defined as the maximum diameter for size dependent processes and aggregation and thus it doesn't describe the maximum diameter of all the particles in the size spectrum. In our revised manuscript, we defined this parameter more clearly in our methods and discussion.

However, the simulated abundance of large particles with a diameter > 4 cm is low and the ability of recording or collecting these large particles is limited, as e.g. the upper size limit of particles that can be measured by the UVP 5 is 26 mm. According to our model assumption, the particle distribution always covers the entire size range from an individual small particle to infinity. (Please note that in our model concept very large particles are extremely sparse, and, because of density decreasing with increasing size, consist almost entirely of water.) Large values of $b$ are associated with a pronounced dominance of small particles and low sinking speeds. As described above, there was a mistake in computing effective sinking speeds in the original manuscript, leading to diagnosed sinking speeds about a factor 16 too high. Although the diagnosed sinking speed is now corrected and generally agrees with previous observations, the particle length scale $b$ is, in regions such as the subtropical gyres, still higher than in observations. Especially in the subtropical gyres, we find a too steep particle size spectrum i.e. too many small particles. Thus, more processes affecting the particle size spectrum, e.g. vertical migration of zooplankton or particle breakup in the deeper ocean, might be necessary. Moreover, it should be noted that our particle length scale, $b$, is calculated by a simple regression using the log-transformed flux and depth. As the aggregation model shows an increase of average sinking speed to a depth of 1,000 m, the calculated particle flux length covers a vertical range of 100 to 1,000 m and thus does not necessarily correspond to the observed depth ranges. However, Marsay et al. (2015) showed that the considered depth range seems to be important for the comparison of the particle flux length. A mismatch of considered depth ranges can thus constitute a potential factor for deviations of $b$ values in our model compared to observations.

We now have extended the discussion on the divergence of simulated and observed particle flux length scales, and on potential processes that might explain this divergence.

Particle size distribution (slope): There is no comparison of the model size spectrum (slope at least) to actual in-situ measurements of particle size distribution which are increasingly available in the literature (ex. Kiko et al., 2017). I would have like to see this comparison in the paper to present evidence that the dynamic of particle aggregation is well capture by the model before to perform any sensitivity analysis.

Thank you for your comment. Please find in our new Figure S5 the number of particles with a size range of 0.14 to 16.88 mm, equal to the range of Kiko et al. (2017). The comparison to the observed transect in the Atlantic equatorial region of Kiko et al. (2017) (in their Figure 1) shows an underestimation of particle concentrations in our model in the surface layer as well as over the full water column. As the model calibration on observed particle data is currently underway, we hope to further improve the fit of particles between model and observations. In our revised manuscript, we compared our model data with the observations by Kiko et al. (2017).

Specific comments

Page 3 L 15 and L 20: Are you referring to Marsay et al., 2015 and Guidi et al., 2015 or Henson et al., 2015 and Marsay et al., 2015 as stated?

Thank you for this comment. Marsay et al. (2015), Henson et al. (2015) and Guidi et al. (2015) showed different patterns regarding the $b$. While Henson et al. (2015) and Guidi et al. (2015) showed similar patterns – although Guidi et al. (2015) presented a more regionalised $b$ – Marsay et al. (2015) found a completely different pattern. We have extended the discussion on regional variations of $b$ by including references to Guidi et al. (2015).

"4. Can the assumptions inherent in the model confirm either of the spatial particle flux length scale maps proposed by Marsay et al. (2015) or Henson et al. (2015) and Guidi et al. (2015)? […] We finally examine and discuss derived maps of particle flux length scales against the background of maps derived from observed quantities (Henson et al., 2015; Marsay et al., 2015; Guidi et al., 2015)."

Table 1 is very hard to go through even though very informative. Representing the 4- last column with a clustergram (heatmap) could help to cluster simulations that present similar outcomes.

Thank you for your suggestion, which is very helpful. We now clustered the last four columns of Table 1 as a heatmap ranging from yellow (high fit to observation) to red (low fit to observations).

*List of changes in*

**The effect of marine aggregate parameterisations on nutrients and oxygen minimum zones in a global biogeochemical model**

Daniela Niemeyer[1], Iris Kriest[1], Andreas Oschlies[1]

[1]GEOMAR Helmholtz-Zentrum für Ozeanforschung Kiel, Düsternbrooker Weg 20, 24105 Kiel, Germany

*Correspondence to: Daniela Niemeyer (dniemeyer@geomar.de)*

**title:** The effect of marine aggregate parameterisations on nutrients and oxygen minimum zones in a global biogeochemical model

**p. 2, l. 7:** The biological carbon pump can be subdivided into three components: production of organic matter and biominerals in the euphotic surface layer, particle export into the ocean interior and finally their decomposition in the water column and on the sea floor (Le Moigne et al., 2013).

**p. 2, l. 13-14:** Recent studies suggest conflicting evidence with regard to the spatial variation of the particle flux length scale
(Guidi et al., 2015; Marsay et al., 2015), which may again be influenced by the methodology of estimating the particle flux profile and thus the potential sensitivity to the considered depth (Marsay et al., 2015).

**p. 3, l. 23:** 4. Can the assumptions inherent in the model confirm either of the spatial particle flux length scale maps proposed by Marsay et al. (2015) or Henson et al. (2015) and Guidi et al. (2015)?

**p. 4, l. 29:** With constant remineralisation rate $r$, the particle flux can thus be described by $F(z) \propto z^{-b}$ with $b = \frac{r}{a}$ (Kriest and Oschlies, 2008), and is therefore (for constant $r$, e.g. in a fully oxic water column) comparable to the common power-law description of observed particles fluxes (Martin et al., 1987).

**p. 5, l. 16-17:** The exponent for the relationship between size and mass is set to $\zeta = 1.62$, as proposed for marine aggregates in Kriest (2002), which is in line with more recent findings (Burd and Jackson, 2009; Jouandet et al., 2014).

**p. 5, l. 21-22:** To prevent instabilities at very large sinking speeds (very flat size distributions), as in Kriest and Evans (2000) and Kriest (2002) we restrict the size dependency of sinking and aggregation to a maximum diameter of $D_L$. Beyond $D_L$,
these processes do not vary with particle size any more.

**p. 8, l. 22-23:** To investigate, if, and how, the model reproduced observed maps of the particle flux length scale, $b$, that relates particle flux and depth via $F(z) \propto z^{-b}$ and derived from data by Marsay et al. (2015) and Guidi et al. (2015), we log-transformed $F(z)$, the simulated, annual average flux of particulate organic matter as a function of depth and carried out a
linear regression of these values.

**p. 9, l. 13-14:** Regions with particularly low diagnosed *b* values (< 0.2) result either from decreased remineralisation in OMZs (e.g. eastern tropical Pacific OMZ) or are found in areas of deep mixing (in the model mainly high latitudes or western boundary currents), where vertical mixing increases the inferred particle flux length scales.

**p. 9, l. 20-21:** In contrast, when deriving the particle flux length scale from a similar model but with oxygen-independent remineralisation (Kriest and Oschlies, 2013), we find a *b* close to the prescribed *b* value of 0.858 (Fig. S1).

**p. 9, l. 30-31:** Although the spatial pattern of export rates is similar for both model simulations with and without aggregation, ECCO1.0* shows a 1.6-fold higher global mean export rate (10.1 mmol P m$^{-2}$ a$^{-1}$) than noAgg$^{ECCO1.0}$ (6.1 mmol P m$^{-2}$ a$^{-1}$).

[revised manuscript text omitted]

**p. 14, l. 21-23:** We therefore suggest, that the difference in improving the representation of OMZs between northern and southern hemisphere is more affected by physics than by biology.

[revised manuscript text omitted]